# Shall We Focus on the Eosinophil to Guide Treatment with Systemic Corticosteroids during Acute Exacerbations of Chronic Obstructive Pulmonary Disease (COPD)? CON

**DOI:** 10.3390/medsci6020049

**Published:** 2018-06-08

**Authors:** Pedro J Marcos, José Luis López-Campos

**Affiliations:** 1Dirección de Procesos Asistenciales, Servicio de Neumología, Instituto de Investigación Biomédica de A Coruña (INIBIC), Complejo Hospitalario Universitario de A Coruña (CHUAC), Estructura Organizativa de Xerencia de Xestión Integrada (EOXI) de A Coruña Sergas, Universidade da Coruña (UDC),15006 A Coruña, Spain; 2Unidad Médico-Quirúrgica de Enfermedades Respiratorias, Instituto de Biomedicina de Sevilla (IBiS), Hospital Universitario Virgen del Rocío/Universidad de Sevilla, 41013 Seville, Spain; lcampos@separ.es; 3Centro de Investigación Biomédica en Red de Enfermedades Respiratorias (CIBERES), Instituto de Salud Carlos III, 28029 Madrid, Spain

**Keywords:** Chronic obstructive pulmonary disease (COPD), acute exacerbation, eosinophil, precision medicine, systemic corticosteroids

## Abstract

The employment of systemic corticosteroids in the treatment of acute exacerbations of chronic obstructive pulmonary disease (COPD) has been shown to improve airway limitation, decrease treatment failure and risk of relapse, and may improve symptoms in addition to decreasing the length of hospital stay. Nowadays, all clinical guidelines recommend systemic corticosteroids to treat moderate or severe COPD exacerbations. However, their use is associated with potential side effects, mainly hyperglycemia. In the era of precision medicine, the possibility of employing blood eosinophil count has emerged as a potential way of optimizing therapy. Issues regarding the intra-individual variability of blood eosinophil count determination, a lack of clear data regarding the real prevalence of eosinophilic acute exacerbations, the fact that previously published studies have demonstrated the benefit of systemic corticosteroids irrespective of eosinophil levels, and especially the fact that there is only one well-designed study justifying this approach have led us to think that we are not ready to use eosinophil count to guide treatment with systemic corticosteroids during acute exacerbations of COPD.

## 1. Introduction

Chronic obstructive pulmonary disease (COPD) is one of the leading causes of morbidity and mortality worldwide [1]. The course of COPD is characterized by episodes of increased symptoms known as exacerbations that, when severe enough, regularly require additional treatments. Exacerbations contribute to a long-term decline in lung function [2], reduced physical activity [3] and also have a significant and long-lasting effect on quality of life [4]. Patients with repeated exacerbations or with exacerbations that require hospitalization have been associated with an increased risk of morbidity and mortality [2,5,6]. Although those with less severe exacerbations are often managed as outpatients, COPD exacerbations may require hospitalization. Although the treatment of exacerbations is a significant contributor to the economic burden of COPD [7], COPD hospitalization is what most contributes to the global cost of the disease [8,9].

## 2. Systemic Corticosteroids in COPD Exacerbation

Chronic obstructive pulmonary disease exacerbation treatment aims to minimize the impact of the current exacerbation and prevent the development of subsequent events. Most COPD exacerbations can be safely managed in the outpatient setting. The main intervention consists of optimizing bronchodilation by increasing the dose and frequency of bronchodilators [10]. The other two key points are antibiotics and systemic corticosteroid treatment.

Although previously published data already suggested the potential benefit of systemic corticosteroids when treating patients with an acute exacerbation of COPD [11,12], it was the publication of a randomized trial by Niewoehner et al. [13] in 1999 which highlighted the importance of this treatment. They compared two regimens of systemic corticosteroids (SC, 2 or 8 weeks) versus a placebo and demonstrated that treatment with SC resulted in improved clinical outcomes (less treatment failure, shorter hospital stays and better lung function). It was also observed that the maximum benefit was obtained in the first two weeks of treatment. Based on these and new results [14], systemic corticosteroids were cemented as a cornerstone of therapy for these patients.

Some years later, Aaron et al. [15] studied the effectiveness of oral prednisone in reducing the risks of relapse after an outpatient exacerbation of COPD. They selected patients discharged from the emergency room who were randomized to receive 10 days of treatment of prednisone or placebo. They found a lower rate of relapse at 30 days (27% vs. 43%; *p* = 0.05), longer time spent relapse-free at 30 days (*p* = 0.04) and a non-significant reduction in hospitalizations for COPD (11% vs. 21%; *p* = 0.11). They also found that after 10 days of treatment, patients treated with systemic corticosteroids had greater improvements in lung function, less dyspnea and better quality of life based on the determinations of the chronic respiratory disease questionnaire (CRQ).

Since those cornerstone studies, new data has reinforced the effect of the treatment (sometimes being compared with steroid nebulized therapy [14,16]) and another studies even have questioned the effect in new hospital settings, like the intensive care unit (ICU) [17]. A Cochrane Review published in 2014 confirmed the benefit of systemic corticosteroids in preventing treatment failure versus placebo (odds ratio (OR) 0.49; 95% confidence interval (CI), 0.35–0.67) [18] with high quality evidence resulting in a number of patients needed to treat to avoid one treatment failure of nine. There was moderate-quality evidence for a lower rate of relapse by one month for treatment with systemic corticosteroids (hazard ratio (HR) 0.78; 95% CI 0.63 to 0.97), better lung function, especially in the first three days (mean difference (MD) 140 mL; 95% CI, 90 to 200), and shorter hospitalization for the inpatient (MD -−1.22 days; 95% CI, −2.26 to −0.18). However, mortality up to 30 days was not reduced by treatment with systemic corticosteroids compared with the control in 12 studies (*n* = 1319; OR 1.00; 95% CI, 0.60 to 1.66), and the likelihood of adverse events, particularly hyperglycemia, increased with corticosteroid (CS) treatment (OR 2.33; 95% CI, 1.59 to 3.43), with a number of patients needed to harm of 6 (95% CI 4 to 10).

The last robust and well-designed study regarding the impact of SC on COPD exacerbations was perhaps the “Reduce trial”, published by Leuppi et al. in JAMA in 2013 [19]. In this study, patients arriving at the emergency department with acute exacerbations of COPD were randomized in order to detect whether five-day treatment with systemic glucocorticoids was non-inferior to 14-day treatment with regard to re-exacerbation within six months of follow-up. The study was positive, demonstrating non-inferior results with the short course, although with significantly reduced glucocorticoid exposure. A review of the most relevant studies regarding the efficacy of systemic corticosteroids can be seen in Table 1.

### Guideline Recommendations

Nowadays, all clinical guidelines recommend systemic corticosteroids to treat moderate or severe COPD exacerbations. Recommendations regarding this issue have been changing in parallel to the development of new studies (Table 2). Systemic corticosteroids treatment was already recommended in the first Global Initiative for Chronic Obstructive Lung Disease (GOLD) workshop summary, published in 2001. Systemic corticosteroids were recommended as an addition to bronchodilator therapy, especially in the hospital management of acute exacerbations of COPD with level A evidence. There were doubts about the exact dosage, even when it was highlighted that high doses were associated with a significant risk of side effects. Finally, 30 to 40 mg of oral prednisolone daily for 10 to 14 days was considered a “reasonable” compromise between efficacy and safety. This recommendation was based on level D evidence, pointing out that prolonged treatment does not result in greater efficacy and increases the risk of side effects.

During subsequent updates to the GOLD Guidelines, evidence regarding the dosage and duration was progressively changed from higher to lower doses and duration. As a result, in GOLD 2006, there was level C evidence for treatment with 30–40 mg of prednisone a day for seven to ten days, and in GOLD 2014, after the publication of the REDUCE (reduction in the use of corticosteroids in exacerbated COPD) [19] results, the recommendation changed to treatment for five days with a dosage of 40 mg a day, with B level evidence. The 2014 update was the first time that GOLD suggested that nebulized corticosteroids (budesonide) alone may be an alternative to oral corticosteroids in the treatment of exacerbations. The last prominent GOLD recommendations regarding treatment with systemic corticosteroids were published in the 2017 edition, suggesting for the first time that systemic corticosteroids may be less efficacious in treating acute COPD exacerbations in patients with lower blood eosinophil levels.

## 3. The Need for Phenotyping Exacerbations and the Potential of Eosinophils

### 3.1. Precision Medicine and COPD Exacerbations

In the current era of precision medicine [23], COPD is increasingly recognized as being comprised of various phenotypes [24,25]. Phenotyping has identified options for targeted therapy, aiming to treat each patient according to his or her own characteristics [26]. Biomarkers that could predict treatment responses to anti-inflammatory drugs may be useful in optimizing the benefit versus risk ratio. There has been more interest regarding the possibility of phenotyping exacerbations since the 2011 publication by Bafadhel et al. [27], in which they showed that there were biologic COPD exacerbation clusters that are clinically indistinguishable and that biomarkers can be used to identify specific clinical phenotypes during exacerbations of COPD. In this study, authors could differentiate four distinct biologic exacerbation clusters: bacterial, viral, pauci-inflammatory and eosinophilic. Interestingly, the four different exacerbation types were the result of a mathematical model based on cluster and principal component analyses, and the crude data presented a clear overlap between the different types. Later that year, Gao et al. [28] also identified four exacerbation types according to cell distribution in sputum. After a lifetime of knowing but not noticing its potential, the eosinophil, specifically the blood count, has now taken center stage as the most clinically accessible biomarker to aid in the management of COPD.

### 3.2. Eosinophilia Associated with COPD

Ever since studies in the early nineties [29] suggested that exacerbations of chronic bronchitis were associated with marked airway eosinophilia, eosinophilic airway inflammation has been classically associated with asthma. Since airway eosinophilia is a reliable predictor of responsiveness to inhaled [30] and systemic corticosteroids treatment [31,32,33], its determination seems crucial for better management of these patients. 

Sputum eosinophil count has been postulated as the best non-invasive way of detecting eosinophilic airway inflammation. However, its measurement has some intrinsic problems including moderate repeatability (intra-class coefficient of 0.49 at 12 weeks [34]) and primarily because it is time consuming, requires certain expertise, and because some patients do not provide adequate samples for analysis or do not provide point-of-care results [35]. As a result, blood eosinophil count appears to be a more practical approach. There has been significant interest in the role of blood eosinophil count in COPD, especially in predicting responses to inhaled corticosteroid therapy exacerbation rates, mostly from the studies showing that the higher the blood eosinophil count, the greater the benefit in reducing exacerbations [36,37].

### 3.3. Blood Eosinophil-Guided Systemic Corticosteroids for Exacerbations of COPD, the Evidence to Date

The predictive role of peripheral blood eosinophil counts as a surrogate biomarker of eosinophilic airway inflammation as a means to direct therapy during exacerbations was recently hypothesized [27]. Some data supporting this approach are that treatment responses to systemic corticosteroids are heterogeneous or do not impact tough clinical outcomes like mortality [18]. However, one of the most important concerns regarding this treatment it is that systemic corticosteroids treatment is associated with significant side effects as hyperglycemia, deep vein thrombosis, weight gain, osteoporosis, infections or neuropsychiatric disorders [13,38,39]). Even short oral corticosteroids courses have been associated with those side effects [39], and the effect is added when patients receive multiple steroids courses when suffering relapsing processes. All of these questions have made some authors to have focused on eosinophil as the cornerstone to guide COPD exacerbation treatment.

Reading carefully some studies regarding this issue, it seems that there is some evidence between patients who responded very well to systemic steroids and another who did not. For example, in the Aaron study [15] the absolute reduction in risk of relapse at 30 days was 16% with a *p* value of 0.05, but 95% confidence interval was between 0 and 32% or the absolute reduction in risk of hospitalization was 10%, but again with a CI from −2 to 22. So, can eosinophil be the biomarker that could help us to discriminate potential “responders”? 

To date, only one study has shown that a biomarker-directed strategy (eosinophil count) can be safely used with good clinical results to guide prescription in COPD exacerbations [40]. The Bafadhel group [39] selected 164 patients, 109 of whom suffered an exacerbation and were randomized to a standard arm, where they received systemic corticosteroids for two weeks, or a biomarker-directed arm, where only patients with a blood eosinophil count of ≥2% received systemic corticosteroids while the others received placebo. The main outcome of the study was the non-inferiority in health status using the chronic respiratory questionnaire (CRQ), the equivalence of the proportion of treatment failure, and evidence of a reduction in CS exposure in the biomarker-directed study group. The results of the study were positive since CRQ improvement in the standard arm was similar to the biomarker-directed group (0.8 vs. 1.1, MD, 0.3; CI 95% 0.0–0.06; *p* = 0.05) and there were no differences in treatment failure (13% vs. 5%; 95% CI, −1 to 16; *p* = 0.07). There was also less exposure to systemic corticosteroids in the biomarker study group since 49% of the exacerbations did not receive systemic corticosteroids. Furthermore, it was found that in biomarker-negative exacerbations, those patients treated with systemic corticosteroids had 15% treatment failures compared with only 2% of those given placebo (*p* = 0.04).

The same group completed a further analysis [41] of randomized controlled studies comparing outcomes for patients with COPD exacerbations treated with prednisolone vs. non-prednisolone (placebo or equivalent). Of just six studies which fulfilled the criteria, only three of them had blood eosinophil count available at the onset of the exacerbation. After analyzing a total of 243 patients (95 patients with 107 exacerbations in the non-prednisolone group vs. 148 patients with 193 exacerbations in the prednisolone group), the study showed that patients presenting with an acute COPD exacerbation and a peripheral eosinophil blood count ≥2% had a significantly reduced treatment failure rate compared with the placebo group (20% vs. 45%; *p* < 0.001).

## 4. Are We Ready to Use Eosinophil Count to Guide Treatment with Systemic Corticosteroids During Acute Exacerbations of COPD? Not Yet

There are some concerns regarding this question which should better be addressed before taking the step to treat patients in this new way.

### 4.1. Can Eosinophils be Trusted in Clinical Practice

Moving on from the role sputum eosinophilia plays in clinical practice due to the problems explained above, there are concerns regarding the use of blood count eosinophilia in the clinical setting. Although it has been proven that there is good agreement between two blood eosinophil count measurements of over a median of 28 days (Ri, intra-class correlation coefficient = 0.8; 95% CI: 0.66–0.88; *p* < 0.001) or even over 1 year (Ri, 0.64), these results represent an average of group results, and when managing individual patients and making decisions based on these results in clinical practice, the benefit may not be so obvious. There is significant intra-subject variability with the peripheral blood eosinophil test, and there are a number of factors that contribute to within-subject variability of blood eosinophil measurements, particularly those that lower the count and could lead to a false negative test [42]. Blood eosinophil counts have been described to vary with exercise, food or medications. There is also a significant diurnal variability of blood eosinophil count with peak levels recorded around midnight and lowest levels at midday and a within-subject biological variation in hourly eosinophil count of 20.9% has been described. Furthermore, although a good correlation between sputum and eosinophilia has recently been suggested, a recent major study based on the SPIROMICS cohort analysis found that blood eosinophils alone showed a significant but weak association with sputum eosinophil counts (receiver operating characteristic area under the curve of 0·64, *p* < 0.0001) with a high false-discovery rate of 72%. It has also been shown to not be a reliable biomarker for COPD severity, exacerbations or sputum eosinophils [43]. This data needs to be taken into account, since SPIROMICS is the largest prospective study to date to have investigated sputum and blood eosinophil counts and their association with clinical outcomes in a population at low risk of exacerbation [44]. Furthermore, there are subtypes of eosinophils that are indistinguishable with the usual hematological stains.

### 4.2. How Many Patients Could Benefit from This Approach

It is not clear what the optimal cut-off point is for blood eosinophils. In the Bafadhel study [27], a cut-off at 2% of blood eosinophil counts distinguishes between eosinophilic and non-eosinophilic COPD exacerbation with a 90% sensitivity. Thus, most studies have adopted this cut-off value. However, even if we take an eosinophil level of 2% as valid, there are not many studies that have investigated the prevalence of eosinophilia on top of an exacerbation. Most of the data (Table 3) came from retrospective studies, with a wide range of prevalence (9.6 to 40%), and different cut-off points (>2%, ≥2%), absolute number of eosinophils, etc. [45,46,47,48,49]. The fact that the eosinophil count is known to fall by >50% within the first four hours following systemic corticosteroids administration [50] and the retrospective nature of these studies highlighted a potential confounding bias arising from the real prevalence of systemic corticosteroids administration in severe eosinophilic COPD. Ultimately, there really isn’t any data regarding the real role of the eosinophil on community exacerbations of COPD.

### 4.3. Did Previous Studies Supporting the Use of Systemic Corticosteroids Measure the Eosinophil Count

There is no doubt regarding the beneficial results of employing systemic corticosteroids to treat COPD exacerbations [18]. Most of the papers that justified their use did not show results regarding the eosinophil count, at least the most robust ones [13,15,19,51]. We do not know if it is because they did not collect the data or they simply did not show it. The point is that those studies demonstrated a positive outcome for systemic corticosteroids independent of eosinophil count levels and, based on their randomized designs, it is expected that eosinophilic and non-eosinophilic patients had been properly distributed between the study arms.

Interestingly, for treatment selection we would like to have a highly predictive negative value to discard candidates or, the other way around, to have a highly positive predictive value to select candidates for therapy. 2% is a low limit so probably the philosophy behind is that of selecting non-responders. In any case, when a higher limit is defined we will then always have a level of uncertainty with middle values, which allows this biomarker to be used in a selection of patients.

### 4.4. Any Light at the End of the Tunnel

The 2017 GOLD guidelines recommendation states that, “Recent studies suggest that glucocorticoids may be less efficacious in treating acute COPD exacerbations in patients with lower levels of blood eosinophils.” To date, it is only based on the 109 patients and 166 exacerbation events in the Bafadhel study [40] we mentioned before, and it appears that in order to make such a strong recommendation leading to a change in the clinical practice in so many patients, further studies and evidence in more patients would be necessary. Although it is a very well-designed study, it does have some limitations (in addition to sample size) that have to be pointed out which could question the recommendation. Firstly, only 10 patients in that cohort suffered from a severe exacerbation, so the application of the eosinophil count to guide treatment with systemic corticosteroids during acute exacerbations of COPD that require hospitalization is not clear and would need more data to be clearly recommended. Secondly, all patients in the study received antibiotics. Although this methodology could be justified in order to replicate the methods employed in the systemic corticosteroids studies where most of the patients received also antibiotics, we would like to highlight that in real clinical setting antibiotic decisions are based on Anthonisen criteria in moderate exacerbations. Thirdly, all of the patients that participated in the study had to have a prior history of exacerbations. We do not know if this approach could be applied to patients with infrequent exacerbations since we do not know if that characteristic could be related to a different therapeutic response. Finally, there is a methodological concern due to the fact that this study applied randomization by minimization which, while a good approach for randomized patients within a small sample size, also has some limitations [52]. Compared with a simple randomization where the allocation of future patients to a trial cannot be predicted, minimization has the disadvantage that, in certain cases, the next allocation can be predicted with certainty with knowledge of the characteristics of earlier patients. There is therefore a potential for selection bias, which can affect the validity of a trial’s results. Even knowing which allocation is more likely to occur next can result in selection bias.

Although no new studies have been published, there is an ongoing, multi-center, randomized, controlled, open-label trial [53] which will help to better address this question. Authors will randomize over 300 patients to test if standard care compared to eosinophil-guided systemic corticosteroids -sparing therapy is non-inferior regarding length of hospital stay (primary outcome). This is expected to answer the uncertainties of Bafadhel´s study as, because patients of the study are admitted to the hospital, patients with and without history of previous exacerbations were included and it has a bigger sample size.

## 5. Is it Realistic for These Recommendations to be Implemented in Clinical Practice in the Short Term

Although the GOLD general recommendation of employing short-term systemic corticosteroids regimens to treat COPD acute exacerbations were first recommended in 2006, when a regimen of 30–40 mg of prednisone was recommended for 7–10 days, and this recommendation was re-adapted in GOLD 2014 based on the REDUCE Trial [19] to 40 mg a day for five days, the application in clinical practice is lacking [37,54,55,56,57,58]. Our group recently published a study where we detected that most of the patients admitted to the hospital with a COPD exacerbation did not receive systemic corticosteroids therapy according to guidelines, this patients received more dosages and more days than recommended, and this was associated with a prolonged hospital stay [54]. As a result, it is a long road of unknown duration from the time Guidelines recommend a change until it is generally accepted and employed.

Another question that has to be taken into account is that, by definition, a blood sample is required. This does not pose any problem in the patients who are admitted to the hospital or who consult an emergency service. However, this can be a problem in outpatient clinics, and specially in primary care where there is usually a large volume of patients and urgent analysis cannot always be done, showing a probably lack of potential application as a practical test of this approach in this setting. Even more, the new approach suggested by the study by Sivapalan et al. [53], where the discontinuation of systemic corticosteroids would be based on a daily blood eosinophil test analysis, argues that this approach could go against clinical practice in the hospital setting.

Although nowadays most COPD patients who suffer from severe COPD exacerbations are elderly people with significant comorbidity and are likely have decreased immunity [59], systemic steroids have been associated with positive outcomes. 

## 6. Conclusions

Systemic corticosteroids have been shown to improve clinical outcomes in the treatment of COPD exacerbations. However, their use is associated with potential side effects. In the era of precision medicine, the possibility of employing blood eosinophil count has appeared as a potential way to optimize therapy. Nevertheless, there are few clinical trials focused on properly addressing the question and more clinical data is needed.

## Figures and Tables

**Table 1 medsci-06-00049-t001:** Most relevant studies regarding the efficacy of systemic corticosteroids (SC).

Author	*n*	Setting	Regimen	Cumulated Dosage (Prednisone, mg)	SC Total Days
Bullard [12]	138	Emergency Department	Arm 1: 100 mg IV hydrocortisone/4 h x 96 h + 4 days oral prednisone 40 mg (5–8 days)	640–760	8
			Arm 2: Placebo	0	0
Niewoehner [13]	147	Inpatient	Arm 1: Methylprednisolone 125 mg/6 h for 72 h, followed by oral prednisone: 60 mg on study days 4 through 7, 40 mg on days 8 through 11, 20 mg on days 12 through 43, 10 mg on days 44 through 50, and 5 mg on days 51 through 57	2985	57
			Arm 2: Methylprednisolone 125 mg/6 h for 72 h, followed by oral prednisone: 60 mg on days 4 through 7, 40 mg on days 8 through 11, and 20 mg on days 12 through 15, and placebo from study days 16 through 57	2355	15
			Arm 3: Placebo	0	0
Maltais [14]	199	Inpatient	Arm 1: Prednisone 30 mg/12 h	210	7
			Arm 2: Nebulized budesonide 2 mg/6 h for 72 h, followed 2000 mcg/d of inhaled budesonide	0	0
			Arm 3: Placebo	0	0
Aaron [15]	147	Outpatient	Arm 1: 40 mg oral prednisone/day	400	10
			Arm 2: Placebo	0	0
Gunen [16]	159	Inpatient	Arm 1: IV prednisolone 40 mg/day for days 1–15 if not discharged, oral methylprednisolone 32 mg/day for days 11–15 if discharged	600	15
			Arm 2: Nebulized budesonide (1500 mg quater in die [q.i.d])	0	0
			Arm 3: no inhaled SC	0	0
Leuppi [19]	314	Inpatient/Outpatient	Arm 1: Methylprednisolone 40 mg on day 1, 40 mg/day days 2 to 5 + placebo days 5–14.	250	5
			Arm 2: Methylprednisolone 40 mg on day 1, 40 mg/day days 2 to 14	610	14
Abroug [17]	217	Intensive care unit (ICU)	Arm 1: Prednisone 1 mg/kg daily until discharge	Weight-based	Maximum 10
			Arm 2: Usual care	0	0

**Table 2 medsci-06-00049-t002:** Global Initiative for Chronic Obstructive Lung Disease (GOLD) recommendations regarding the treatment of acute exacerbations of COPD with SC.

Guideline	Drug	Dosage	Duration	Level of Recommendation	Note
GOLD 2001 [20]	Prednisolone	30–40 mg/d	10–14 days	D	The exact dose that should be given is not known, but high doses are associated with a significant risk of side effects Prolonged treatment does not result in a greater efficacy and increases the risk of side effects
GOLD 2006 [21]	Prednisolone	30–40 mg prednisone/d	7–10 days	C	
GOLD 2014 [1]	Prednisone	40 mg/d	5 days	B	Nebulized budesonide alone may be an alternative (although more expensive) to oral corticosteroids in the treatment of exacerbations
GOLD 2017 [22]	Prednisone	40 mg/d	5 days	B	Recent studies suggest that glucocorticoids may be less efficacious in treating acute chronic obstructive pulmonary disease (COPD) exacerbations in patients with lower blood eosinophil levels

**Table 3 medsci-06-00049-t003:** Prevalence of eosinophilia in severe COPD exacerbations.

	n	Design	Cut-off		
			Eosinophils >2%	Eosinophils ≥2%	Eosinophils >300 cells/µL
Hasegawa [45]	3084	Retrospective		40%	17%
Salturk [46]	647	Retrospective	9.6%		
Duman [44]	1704	Retrospective	20.6%		
Serafino-Agrusa [47]	132	Retrospective		15.1%	
Couillard [48] *	167	Retrospective		32.9%	

* Cut-off point defined as: Eosinophils ≥2% and/or ≥200 cells/µL of the total leukocyte count.

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
