# Peer review of "Shall We Focus on the Eosinophil to Guide Treatment with Systemic Corticosteroids during Acute Exacerbations of Chronic Obstructive Pulmonary Disease (COPD)? CON"

_medsci, 2018, doi:10.3390/medsci6020049_

Round 1

Reviewer 1 Report

This is an excellent review of a topical subject. I think readers of the journal will find it interesting.I have only minor suggestions:

There is some more up-to-date information on potential side effects of short course oral steroids (see BMJ Open 2017).This could be discussed.

Please do not use SC as an abbreviation for systemic corticosteroids. This  is widely understood to refer to subcutaneous. I suggest OCS or no abbreviation.

The optimum cutpoint could be discussed more. Isn't this a situation where you want a very high negative predictive value? This means that few potentially responsive patients would be dened treatment. I think this is where the 2% value came from.

Author Response

This is an excellent review of a topical subject. I think readers of the journal will find it interesting. I have only minor suggestions:

There is some more up-to-date information on potential side effects of short course oral steroids (see BMJ Open 2017).This could be discussed.

- Thanks for the suggestion, we have added more information regarding this and added one more references (Lines 153 to 158).

Please do not use SC as an abbreviation for systemic corticosteroids. This  is widely understood to refer to subcutaneous. I suggest OCS or no abbreviation.

- Since the concept of systemic corticosteroids is broader than oral corticosteroids, we have deleted the abbreviation all along the manuscript.

The optimum cutpoint could be discussed more. Isn't this a situation where you want a very high negative predictive value? This means that few potentially responsive patients would be denied treatment. I think this is where the 2% value came from.

- Thank you for the comentary, We agree with the reviewer. For treatment selection we would like to have a highly predictive negative value to discard candidates or, the other way round, to have a highly positive predictive value to select candidates for therapy.2% is a low limit so probably the philosophy behind is that of selecting non-responders. In any case, there will always be a level of uncertainty with middle values, which allows this biomarker to be used in a selection of patients really. We have included this argument in the manuscript, as per the suggestion.

Reviewer 2 Report

This is a well written "con" article for the use of eosinophils to guide oral steroid use in acute exacerbations of COPD, which I enjoyed reading.  I would like the authors to consider the following points:

1. Although the evaluation of studies to support the use of SC in COPD were well presented in general, not all were presented in a manner which truly reflected the data.  For example, the authors say "Since those cornerstone studies, more data has reinforced the effect of the treatment, sometimes being compared with steroid nebulized therapy [14,16] or in new hospital settings, like the ICU [17]".

Actually, the conclusion of the Abroug study is "Prednisone did not improve intensive care unit mortality or patient-centred outcomes in the selected subgroup of COPD patients with severe exacerbation but significantly increased the risk of hyperglycaemia.".  So this trial does not support a treatment effect - rather the opposite.     This needs amending.

2. Even in the SC treated groups, the risk of exacerbation relapse was high and the authors do not consider the potential side effects and adverse events associated with repeated short courses of steroids which include DVT, bone density loss, hyperglycaemia, disturbed mood and sleep, weight gain, etc.  This is not a "risk free" treatment option in patients who exacerbate frequently and there should be some consideration of this.  Just mentioning hyperglycaemia is probably not enough.

3. In the SC studies quoted, there is clear evidence of "responders" and "non-responders".  For example, in the Aaron study (New Eng J Med) the absolute reduction in risk of relapse at 30 days was 16%, but 95% CI were 0-32%, the absolute reduction in risk of hospitalisation was 10% (but CI were -2 to 22), showing some patients had a big response to steroids, and some did not.   This is not a one treatment fits all.

4. So the meat of the con argument is whether the eosinophil is the right biomarker to guide SC treatment. 

I think this question has 3 parts - 

A. Is the eosinophil a good marker of exacerbation risk?  (and the SPIROMICS paper the authors quote states that " Exacerbations requiring corticosteroids treatment were more common in the high versus low sputum eosinophil group (p=0·002)"   but the variability of this marker is well described in this article.   

B.  Is the eosinophil a good marker to predict benefit from SC in AECOPD?  There is some evidence to support this, but more data is needed.

C.  Is the eosinophil a practical test to guide treatment in exacerbations?  Here there are real questions, especially in a primary care, non-hospital setting.   Perhaps these could be separated more clearly.

5. The authors suggest that the Bafadhel study is limited as "all patients in the study received  antibiotics. Since most of the patients were moderate, this approach is difficult to justify in real clinical practice" but most of the SC studies they quote showing the benefit of SC also treat all patients with antibiotics, so is this a fair criticism?

6.  Can the authors provide a reference to support the statement that intra-patient variations of eosinophil counts can be up to 20.9%?

All in all, I really enjoyed this piece of work, but wondered if slightly more balance is needed in places?

Author Response

This is a well written "con" article for the use of eosinophils to guide oral steroid use in acute exacerbations of COPD, which I enjoyed reading.  I would like the authors to consider the following points:

1. Although the evaluation of studies to support the use of SC in COPD were well presented in general, not all were presented in a manner which truly reflected the data.  For example, the authors say "Since those cornerstone studies, more data has reinforced the effect of the treatment, sometimes being compared with steroid nebulized therapy [14,16] or in new hospital settings, like the ICU [17]". Actually, the conclusion of the Abroug study is "Prednisone did not improve intensive care unit mortality or patient-centred outcomes in the selected subgroup of COPD patients with severe exacerbation but significantly increased the risk of hyperglycaemia.".  So this trial does not support a treatment effect - rather the opposite.     This needs amending.

- We agree authors that that paragraph did not reflected the sense authors wanted to give. We have rewritten that paragraph (lines 74 to 76)

2. Even in the SC treated groups, the risk of exacerbation relapse was high and the authors do not consider the potential side effects and adverse events associated with repeated short courses of steroids which include DVT, bone density loss, hyperglycaemia, disturbed mood and sleep, weight gain, etc.  This is not a "risk free" treatment option in patients who exacerbate frequently and there should be some consideration of this.  Just mentioning hyperglycaemia is probably not enough.

- Thanks for the suggestion, we have added more information regarding this and added one more references (Lines 150 to 158).

3. In the SC studies quoted, there is clear evidence of "responders" and "non-responders".  For example, in the Aaron study (New Eng J Med) the absolute reduction in risk of relapse at 30 days was 16%, but 95% CI were 0-32%, the absolute reduction in risk of hospitalisation was 10% (but CI were -2 to 22), showing some patients had a big response to steroids, and some did not.   This is not a one treatment fits all.

- We absolutely agree the reviewer comment. That is why we have incorporated into the manuscript a new paragraph (lines 159-164 regarding this issue.

4. So the meat of the con argument is whether the eosinophil is the right biomarker to guide SC treatment. 

I think this question has 3 parts - 

A. Is the eosinophil a good marker of exacerbation risk?  (and the SPIROMICS paper the authors quote states that " Exacerbations requiring corticosteroids treatment were more common in the high versus low sputum eosinophil group (p=0·002)"   but the variability of this marker is well described in this article.   

- Ok, thanks

B.  Is the eosinophil a good marker to predict benefit from SC in AECOPD?  There is some evidence to support this, but more data is needed.

- Agree.

C.  Is the eosinophil a practical test to guide treatment in exacerbations?  Here there are real questions, especially in a primary care, non-hospital setting.   Perhaps these could be separated more clearly.

- We have added new sentences to reinforce this message (lines 283 to 285)

5. The authors suggest that the Bafadhel study is limited as "all patients in the study received  antibiotics. Since most of the patients were moderate, this approach is difficult to justify in real clinical practice" but most of the SC studies they quote showing the benefit of SC also treat all patients with antibiotics, so is this a fair criticism?

- Thanks for this. We expected this argument could be employed by the pro-authors, however we decided to clarify the issue (lines 251-254).

6.  Can the authors provide a reference to support the statement that intra-patient variations of eosinophil counts can be up to 20.9%?

- We took the value from the paper of the reference #42.

All in all, I really enjoyed this piece of work, but wondered if slightly more balance is needed in places?

We would like to thank the reviewer for the thorough evaluation of the manuscript, the positive comment and the insightful comments that have allowed us to improve it. We have now review it in the light of these comments to try to give it a more balanced view, considering that this is a CON argument paper.